# Application of Metal–Organic Framework in Diagnosis and Treatment of Diabetes

**DOI:** 10.3390/biom12091240

**Published:** 2022-09-05

**Authors:** Qian Gao, Que Bai, Caiyun Zheng, Na Sun, Jinxi Liu, Wenting Chen, Fangfang Hu, Tingli Lu

**Affiliations:** School of Life Sciences, Northwestern Polytechnical University, 127 West Youyi Road, Beilin District, Xi’an 710072, China

**Keywords:** diabetes, metal–organic framework, synthetic, diagnosis of diabetes, sensors, biological materials, wound healing, skin regeneration

## Abstract

Diabetes-related chronic wounds are often accompanied by a poor wound-healing environment such as high glucose, recurrent infections, and inflammation, and standard wound treatments are fairly limited in their ability to heal these wounds. Metal–organic frameworks (MOFs) have been developed to improve therapeutic outcomes due to their ease of engineering, surface functionalization, and therapeutic properties. In this review, we summarize the different synthesis methods of MOFs and conduct a comprehensive review of the latest research progress of MOFs in the treatment of diabetes and its wounds. State-of-the-art in vivo oral hypoglycemic strategies and the in vitro diagnosis of diabetes are enumerated and different antimicrobial strategies (including physical contact, oxidative stress, photothermal, and related ions or ligands) and provascular strategies for the treatment of diabetic wounds are compared. It focuses on the connections and differences between different applications of MOFs as well as possible directions for improvement. Finally, the potential toxicity of MOFs is also an issue that we cannot ignore.

## 1. Introduction

As the largest organ of the human body, the skin has an important impact on the life activities and physiological functions of the human body, including resisting pathogens, sensing the external environment, and regulating body temperature [1]. According to the World Health Organization (WHO), diabetes affects approximately 422 million people worldwide, most of whom live in low- and middle-income countries, and approximately 1.5 million people die of diabetes each year [2]. Both the incidence and prevalence of diabetes have steadily increased over the past few decades. Hyperglycemia, infection, chronic inflammation, poor angiogenesis, and granulation tissue formation are also common problems in diabetic wounds. These factors contribute to a prolonged period of diabetic wound healing, and the resulting chronic wounds may even lead to amputation [3,4]. With projections that up to one third of U.S. adults may develop diabetes by 2050, the impacts and costs associated with chronic diabetic wounds are expected to rise dramatically, creating an urgent need for new treatments to address clinical needs.

Metal–organic frameworks or coordination polymers, or MOFs for short, are highly tunable hybrid materials that are bridged by organic ligands and coordinating metals to form various topological isomeric structures [5]. The formation and morphology of MOFs depend on the types of metal sources and ligands used in the reaction process, and this near-infinitely combinable structural advantage provides opportunities for their surface functionalization. MOFs have been used in fields such as storage and separation [6], biosensing [7], bioimaging [8], and biocatalysis [9] due to their unique crystal structure properties (e.g. large specific surface area, high porosity, and good biocompatibility and biodegradability) and have rapidly gained more attention in the biomedical field due to their targeted delivery, stability, the controlled release of their loaded contents, and other biological properties [10,11] (Figure 1). Through a large number of studies, MOFs have been upgraded from the initial therapeutic delivery and sustained and controlled release to the intelligent direction. These functionalized MOFs can be loaded with therapeutic substances, have programmability, and are more conducive to dealing with complex wounds while achieving coordinated antibacterial, anti-inflammatory, pro-angiogenic, fibroblast generation, and collagen deposition outcomes, effectively improving the safety of treatment [12]. 

Several reviews have summarized the applications of MOFs in water treatment [9,13], battery electrodes [14], and biomedicine [15], but few have systematically organized their applications in diabetes therapy, including the diagnosis and treatment of diabetes and diabetic wound ulcers. In this review, we summarize the synthesis methods of MOFs and the recent progress in the application of MOFs in diabetes and introduce some recent strategies for diabetes diagnosis and drug delivery. This paper focuses on the application of MOFs in the diagnosis and control of blood glucose and its specific mechanism of antibacterial, anti-inflammatory, anti-oxidative stress, and vasogenic application in wound healing. There is also a problem that cannot be avoided in the application process: the toxicity of MOFs nanoparticles. Finally, we will discuss the challenges and opportunities in this field as well as future research directions. We believe that this review will be able to help readers better understand the formation mechanism of different MOF-based porous nanoplatforms and the application progress.

## 2. The Synthesis of MOFs

Different synthetic methods can affect the structure and shape of MOFs and their biological functions [16]. Currently, many different synthetic methods can be used to generate MOFs, depending on the framework and property requirements. Commonly used synthesis methods include hydrothermal/solvothermal synthesis [16,17,18], room-temperature synthesis, microwave-assisted synthesis [19], ultrasonic-assisted synthesis [20,21], mechanochemical synthesis [22], microfluidic synthesis [23,24], biomimetic mineralization [25,26], and so on (Table 1).

### 2.1. Hydrothermal/Solvothermal Synthesis

In the hydrothermal process, water is used as the solvent, and the raw materials are prepared into different solutions, heated to above 100 °C in a hydrothermal kettle, and cooled to room temperature naturally after a certain period of reaction [41]. They are then collected by centrifugation. This method is characterized by generality, simplicity, and low cost. The precise trapping of narrow pores and the stable bonding of vacancies not only simplify the synthesis process of atomically dispersed catalysts but also enable their large-scale preparation at mild temperatures [18]. The solvothermal method is usually based on an oil bath, which is heated to a relatively high temperature for the reaction [42].

### 2.2. Room-Temperature Synthesis

When the loading biological activity is unstable or the sustainability requirements are relatively high, the room-temperature method, also called the one-pot precipitation method, is generally used. When loading a multienzyme system, MOFs can easily immobilize different cargoes, and the substrate channel effect can promote the elimination of enzyme cascade intermediates and improve the catalytic efficiency [21]. The room-temperature method is usually the preferred synthesis method due to the easy realization of reaction conditions and simple operation.

### 2.3. Microwave-assisted Synthesis

The microwave-assisted method can rapidly synthesize nanoporous materials under hydrothermal conditions. Microwave heating increases the energy in the reaction, accelerates the reaction process, and can shorten the reaction time from tens of hours to a few hours or even several minutes [19]. The advantages of this method are that there are no excessive by-products and the high purity and small size of the MOFs obtained [43]. Selecting a solvent with good thermal conductivity is one of the strategies to improve the conversion rate [44]. Microwave-assisted techniques appear to be the best method to rapidly and efficiently prepare MOFs in high purity and yield with controlled size.

### 2.4. Ultrasound-assisted Synthesis

The ultrasound-assisted method, also called the sonochemical method, is beneficial to quickly disperse solutes and speed up the reaction process and is usually used as a method to improve the reaction efficiency [20]. The advantage is that it is more environmentally friendly, and compared with the traditional synthesis method, the synthesis time is greatly reduced [45,46,47]. In general, the resultant nanocrystals are smaller in size than with solvothermal methods.

### 2.5. Mechanochemical Synthesis

Mechanization is the generation of mechanical forces by friction and collision, combining a truncated mixed ligand strategy and defect engineering theory to provide energy for chemical synthesis [48]. A simple, green, and rapid construction strategy was used to harvest gram-scale microporous/mesoporous MOFs in solvent conditions [48,49]. In general, parameters such as the activation solvent, metal-to-ligand ratio, grinding speed, and time will affect the quality of the product.

### 2.6. Microfluidic Synthesis

In this system, the aqueous phase of the metal salt and the organic ligand solution meet at the T-junction through a syringe pump, and then the mixed solution flows together in the form of a laminar flow or liquid droplets in the microfluidic system to achieve the continuous production of MOFs [50,51].

### 2.7. Biomineralization Synthesis

Biomineralization refers to the process of generating inorganic minerals from organisms through the regulation of biological macromolecules. Under certain physical and chemical conditions, proteins, enzymes, and DNA can rapidly induce the formation of protective metal–organic framework coatings by converting ions in solution into solid minerals under the control or influence of biological organic substances [25,26]. Biomimetic mineralization provides unprecedented protection from biological, thermal, and chemical degradation while maintaining biological activity by encapsulating biological macromolecules in porous metal–organic frameworks [52]. He et al. combined a low-cytotoxic and pH-sensitive biodegradable MOF material (zeolitic imidazole framework-8 (ZIF-8)) with insulin (INS), GOx, and catalase (CAT) to form glucose-responsive NPs as functional “cores”, enabling long-term glucose-responsive insulin delivery [40].

## 3. Application of MOFs in Diabetes Diagnosis

There are three main reference values for the diagnosis of diabetes: (1) a fasting plasma glucose greater than or equal to 7.0 mM, (2) a random plasma glucose greater than or equal to 11.1 mM., and (3) an OGTT test two-hour plasma glucose greater than or equal to 11.1 mM. In order to monitor the blood sugar status of patients in real time, sensors with high sensitivity and high selectivity are particularly important. Compared to traditional invasive blood sampling, breath analysis is a noninvasive method suitable for frequent blood glucose monitoring. Acetone (ACE), a ketone produced when the body is forced to use stored fat as its primary source of energy, is now a widely accepted biomarker of diabetes [53,54,55,56]. Similarly, other immunological indicators have also been used as indicators to detect blood glucose levels, for example, glycosylated hemoglobin (HbA1c) [57], INS [58], retinol-binding protein 4 (RBP4) [59], tumor necrosis factor-α (TNF-α) [60], and so on. HbA1c is the free amino group of the N-terminal valine of the hemoglobin beta chain of different carbohydrates formed by glycosylation and is closely related to the fasting blood glucose level in the past 2 to 3 months. Insufficient insulin release or insulin loss in target tissues can also easily lead to the occurrence of diabetes [61,62]. Insulin is involved in regulating blood sugar balance in the body, facilitating the absorption of glucose from fats and proteins. Therefore, it is necessary to develop an efficient and sensitive method for detecting insulin in serum. Xiang et al. found that RBP4 levels are increased when insulin resistance and diabetes occur [63,64]. During diabetes, the level of the inflammatory factor TNF-α is also elevated [60]. In recent years, MOF nanoparticles have been used to enhance the catalytic activity of immunosensors by virtue of their open metal sites and tunable microporous structures [65,66]. Therefore, many MOF-based electrochemiluminescence (ECL) biosensors have been developed for the detection of diabetes, including blood glucose sensors [67,68,69], acetone and isopropanol sensors [70,71,72], and other sensors [57,58,59,60]. 

### 3.1. Glucose Sensor

In previous studies, a novel lanthanide-functionalized metal–organic framework enzyme (L-MOF-enzyme) complex was prepared using a surface adhesion strategy between Eu^3+^@UMOF and glucose oxidase (GOx) [73]. Among them, Eu^3+^@UMOF could be used as a support fixation, drove the cascade reaction to form H_2_O_2_, and acted as a fluorescent center in response to glucose (Glu). This MOF complex enzyme showed better stability and high fluorescence selectivity and sensitivity to Glu (the detection limit was about 0.2 μM). In another study, glutaraldehyde (GA) was used as the connector and glucose oxidase (GOD) was modified to prepare nickel/copper bimetallic organic framework biosensor (GOD-GA-Ni/Cu-MOFs-FET) [67]. Although this sensor did not have good long-term stability, it could be used as a one-off real-time glucose sensor due to its good specific reproducibility and fast response time (Figure 2A). Wang et al. synthesized Co-based two-dimensional (2D) metal azolate framework nanosheets (MAF-5-CoII NS nanozyme) and developed a disposable glucose sensor (MAF-5-CoII NS/SPE sensor) with good redox activity in neutral and alkaline media that could induce the catalytic oxidation of glucose and could be used for the non-enzyme detection of glucose in diluted human plasma samples [74]. The detection limits of the sensor under neutral or alkaline conditions were 0.25 and 0.05 μM, respectively, while its maximum sensitivities were 36.55 and 1361.65 mA/cm^2^/mM. In addition, the copper-based 2D metal–organic framework nanosheets can also simulate the activity of horseradish peroxidase to directly detect H_2_O_2_ and conduct glucose fluorescence sensing through fluorescence sensing coupled with GOx [75] (Figure 2B).

### 3.2. Acetone and Isopropanol Sensors

In general, the ACE concentration in the exhaled breath in diabetic patients is higher than 1.8 ppm, and the concentration in normal people is lower than 0.9 ppm [72,76,77]. Because of the link between diabetes and acetone, doctors recommend that patients check the lever of ketone and blood sugar frequently. In addition, isopropanol (IPA) is regarded as a new biomarker for diabetes. It is an acetone reduced by the reverse reaction of alcohol dehydrogenase [78,79]. Therefore, the indirect monitoring of the amount of biomarker, such as the ACE [80,81] and IPA biomarkers, could help us monitor diabetic blood glucose levels more conveniently. At present, we know that various types of gas sensors have been invented [82,83,84,85], for example, the following MOF-based ACE and IPA sensor.

Due to the low concentrations of ACE and IPA in human breath, they are difficult to measure directly. Therefore, it is critical to develop efficient ACE and IPA enrichment techniques prior to analytical procedures. Yu et al. selected three types of MOFs (ZIF-7, UiO-66, and MOF-5) as adsorbents [6], found UiO-66 was a suitable adsorbent, and successfully applied it to detect the concentrations of ACE and IPA in real breath samples with good sensitivity and high recovery.

Co-doped ZnO NPs were synthesized and exhibited a high response (18.2 at 5 ppm) to trace ACE [72], a fast response/recovery time, a low detection limit of 0.17 ppm, and long-term stability up to 4 months. Combining an MOF with an MOS (metal oxide semiconductor) to form heterostructures is an effective strategy to improve gas sensing performance. For example, a complex of ZnO@ZIF-71(Co) was recently reported to detect acetone gas [86]. This gas sensor exhibits a high response to acetone (5.7 for 0.5 ppm acetone), with a detection limit as low as 50 ppb, a short response time (71 s), and a fast recovery time (53 s) (Figure 3A). 

Postsynthesis modification by the metal-exchange approach can be accomplished by the charge-balanced substitution of cations present in the pores or more complex processes to form co-mixed metal MOF (MM-MOF) materials of multiple metals in a single MOF crystal [87,88]. Some engineered MOFs exhibit excellent luminescent properties [89], exploiting the uncoordinated oxygen atoms of the terephthalate linker present in Zn-BDCs to interact with the Ag cations in a silver nitrate (AgNO_3_) solution, resulting in Zn-BDCs converted to Ag-BDC that exhibited a selective reaction to acetone and a strong green luminescence, with emission quantum yields as high as 22% in the solid-state (powder) form [56]. This green emission is strongly quenched (>90% reduction in the initial emission intensity) in the presence of acetone, making them promising candidates for making fluorescent sensors for diagnosing and monitoring diabetes. Unfortunately, the underlying mechanism of acetone sensing is currently unclear (Figure 3B). In conclusion, luminescent metal–organic frameworks (LMOFs) exhibit excellent photoluminescence properties, high selectivity, anti-interference, high sensitivity, and fast responses but suffer from poor water stability in practical applications. Therefore, methods to improve the water stability of MOFs in follow-up studies are of great significance [90,91,92].

### 3.3. Other Sensors

An antibody-labeled zirconium metal–organic framework/Fe_3_O_4_ (trimethyl chitosan)/gold nanocluster (Zr-MOF/Fe_3_O_4_(TMC)/AuNCs) nanocomposite can selectively capture HbA1c and isolate it from nonglycosylated Hb species and other glycosylated proteins in human blood samples [57]. Disposable Au electrodes were modified with a copper (II) benzene-1,3,5-tricarboxylate (Cu-BTC)/phyllomidazole molecular leaf-like framework (ZIF-L) for insulin detection [58]. The aptamer was easily immobilized on the Cu-BTC/ZIF-L composite by physical adsorption, which promoted the specific interaction between the aptamer and insulin. The linear detection range of insulin based on the Cu-BTC/ZIF-L composite was wide (0.1 pM–5 μM), and the lower detection limit was 0.027 pM. RBP4 has been regarded as an important serological biomarker for type 2 diabetes mellitus (T2DM). Taking advantage of a dual-signal quenching @CNT composite between a luminol@AuPt/ZIF-67 hybrid and MnO_2_, an ECL immunosensor for RBP4 detection is proposed, providing another detection strategy for the early clinical diagnosis of T2DM (Figure 4). Specifically, the multifunctional AuPt/ZIF-67 hybrid has high peroxidase-like nanase activity and loading capacity, which can improve the ECL performance of the luminol-H_2_O_2_ system [59]. The MnO_2_@CNTs composite can effectively quench the amplified initial signal by inhibiting the peroxidase-like activity of luminol@AuPt/ZIF-67 and the ECL-RET strategy. We know that carbon nanotubes (CNTs) have been widely used in the modification of metal oxides and other materials due to their unique nanosize, high specific surface area, superior adsorption effect and multifunctional binding sites [59]. Yola et al. developed a sensor platform on the surface of glassy carbon electrodes (GCEs) using a mixture of thiol-functionalized multiwalled carbon nanotubes (S-MWCNTs) and gold nanoparticles. Then, the captured TNF-α antibody binds to the sensor platform via amino-gold affinity. After TNF-α antibody immobilization, the immunoreaction between AuNPs/S-MWCNTs immobilized with TNF-α primers and TNF-α secondary antibody-conjugated bimetallic Ni/Cu-MOFs were used to prepare a novel voltammetric TNF-α immunosensor [60].

## 4. Application of MOFs in Regulating Blood Glucose

### 4.1. Oral Insulin Delivery

The direct injection of insulin (INS) is currently the only effective treatment for diabetes [93]. INS has a short blood half-life and requires intensive self-management by patients during injection and frequent dose adjustments according to blood glucose level (BGL) monitoring. Long-term injection can lead to local tissue necrosis, nerve damage, and microbial infection [94]. Therefore, it is necessary to develop other types of INS delivery methods. Oral insulin is associated with better patient compliance than injection. However, INS is a pH- and enzyme-sensitive protein. It is well-known that the main challenges of oral INS delivery strategies include, on the one hand, the strongly acidic environment in the stomach and pepsin-mediated degradation as well as digestive-enzyme-mediated degradation in the gut. On the other hand, it is the low permeability of protein drugs across the gut biomembrane that results in low bioavailability [95,96]. Both mucous cells and epithelial cells in the gut act as barriers. Therefore, an optimal oral delivery system should not only protect insulin from degradation in the gastrointestinal environment but also promote its permeability through intestinal epithelial cells. The alkaline environment of the gut may increase the availability of therapeutically active ingredients and can transport them directly to the systemic circulation through the hepatic hilum or the intestinal lymphatic system [97]. 

As early as 1993, Archibald et al. proposed the concept of biomineralization to achieve the high-fidelity replication and functional specificity of biomaterials and biopolymers [98]. The most typical biomineralization product is the formation of MOF complexes. MOFs have emerged as a promising class of proteins by confining proteins within their rigid frameworks with tunable mesoporous structures and high surface area and water stability while achieving high loadings and significantly improving the thermal and chemical stability of encapsulated proteins, functional protein immobilization, and storage materials [99,100]. Through the uniform mesoporous structure, the loading and release of insulin can be flexibly controlled.

The combination of MOFs with other materials (such as hydrogels, microspheres, proteins, etc.) will enable MOFs to have multiple functions. In the latest study, Rojas et al. synthesized CS@MIL-127 as drug delivery systems (DDSs) that exploited the good biocompatibility, large porosity, and excellent stability along the gastrointestinal tract of microporous iron-based nanoMOFs [97]. In addition, the use of the biopolymer chitosan greatly improved the intestinal permeability of MIL-127, and the safety and efficacy of the DDSs were confirmed using C. elegans (Figure 5A). Zhou et al. developed a novel nanocomposite carrier (Ins@MIL100/SDS@MS) based on iron-based MOF nanoparticles (NP) (MIL-100) with high loadings of biodegradable microspheres for oral INS delivery [101]. The microspheres effectively protected MOF NPs from rapid degradation under acidic conditions and could release INS-loaded MOF NPs in simulated intestinal fluid (Figure 5B). Chen et al. used Zr6-based mesoporous MOF NU-1000 as a carrier for INS and achieved the rapid encapsulation and stabilization of INS within 30 min [102]. To further improve the delivery efficiency of INS, Jun et al. designed acid-resistant nMOFs (UiO-68-NH_2_) with external targeting protein (Tf) modification for efficient oral INS administration [103] (Figure 5C). Among them, UiO-68-NH_2_ has high porosity, acid stability, and phosphate ion instability (Figure 5D). Decorating the exterior of UiO-68-NH_2_ with a targeting protein (transferrin) can simultaneously protect insulin from acid and enzymatic degradation, resulting in efficient oral INS delivery. 

### 4.2. Wound Intelligent Insulin Delivery

To achieve a “smart” BGL-dependent insulin release process in vivo, He et al. synthesized a “closed-loop” intravenous insulin delivery system (“core-shell” erythrocyte membrane-encapsulated MOF) that could mimic the dynamics of pancreatic β cells and the secretion of glucose-responsive insulin [40]. Insulin, glucose oxidase (GOx), and catalase (CAT) were co-encapsulated into ZIF-8 nanoparticles (NPs). Among them, an efficient enzyme cascade system (GOx/CAT group), as an optimized glucose response module, can rapidly catalyze the glucose production of gluconic acid to reduce local pH and effectively consume the harmful by-product hydrogen peroxide (H_2_O_2_), inducing pH-sensitive ZIF-8 NP collapse to release insulin. Erythrocyte membranes reduce the elimination of NPs by the immune system, resulting in a steady decline in BGLs and the maintenance of normoglycemia for up to 24 h without hypoglycemia or hyperglycemia spikes. This approach enables the nanoparticles to be intravenously injected and have long-term stability in the blood circulation, enabling long-term INS delivery (the loading efficiency of INS is 20.2–24.8%). Similarly, Rohra et al. used microfluidics to further increase the insulin loading in ZIF-8 to 88% for biosensing applications [51]. Zeolitic Zn^2+^-imidazolate cross-linked scaffold nanoparticles ZIF-8 NMOF were used as “smart” glucose-responsive carriers for controlled drug release. For a better release of INS, Chen et al. chose boric-acid-modified gel as a glucose-triggered self-regulating matrix to release insulin. The conjugation of glucose with boronic acid functional groups generates charged boronic ester residues, resulting in hydrogel swelling and insulin release [104]. In these studies, the primary goal is to increase the bioavailability of cargo, monitoring only drug or MOF components (cations or ligands) without fully considering the potential intestinal crossover of nanoscale MOFs (nanoMOFs) as the entire carrier.

## 5. Distinguishing between Normal Wounds and Diabetic Wounds

### 5.1. Normal Wounds

Wound healing is a complex process that can be divided into several stages, including hemostasis, inflammation, proliferation, and wound remodeling (wound shrinkage). Platelets first start to aggregate and migrate to the site of damaged blood vessels, reducing blood flow. At the same time, platelets release chemokines, and white inflammatory cells such as neutrophils and macrophages are attracted to the injury site. These cells release reactive oxygen species (ROS) and matrix metalloproteinase (MMP)-8, thereby eliminating the open wound site of bacteria and cell debris [105]. The proliferative process generally begins on day 2 after injury with the appearance of granulation tissue. At this time, fibroblasts are responsible for the production and modification of ECM in granulation tissue with the help of matrix metalloproteinases (MMPs). Initially, the ECM rapidly produces a network of type III collagens, which are weaker forms of structural proteins [106]. During the remodeling phase, the replacement of type III collagen with type I collagen results in wound closure and scarring [105].

### 5.2. Diabetic Wound Healing

When any component of the wound healing process is compromised, healing may be delayed. Chronic wounds will develop when a person suffers from an associated disease such as diabetes [107]. Chronic wounds are those that fail to follow the normal wound healing sequence and do not achieve abnormal healing. The treatment of diabetic wounds remains a major clinical challenge due to their complex wound healing environment, characterized by imbalanced inflammatory responses, the over-expression of reactive oxygen species, hyperglycemia, a lack of angiogenesis, and an extremely high risk of bacterial infection [108,109,110]. This wound environment results in chronic and excessive inflammation at the wound site, recurrent infections, and an inability of the dermis and epidermis to respond to regenerative stimuli. The inability of the wound to heal, combined with the high sugar environment, provides a comfortable breeding ground for bacterial growth, further delaying wound healing [111].

Based on the characteristics of the internal environment in diabetic wounds, in recent years the invention of MOF-based nanomaterials was mainly aimed at the problems of repeated infection, unbalanced inflammatory reaction, excessive oxidative stress, and the insufficient angiogenesis caused by hypoxia [96,97,98] by improving wound site infection and blood flow reperfusion to accelerate the wound healing process and restore normal skin function.

## 6. Application of MOFs in Wound Healing

Based on the characteristics of the internal environment in diabetic wounds, in recent years, many porous materials and biodegradable polymers have received extensive attention and demonstrated important applications in wound repair. Compared with traditional inorganic polymers (mesoporous silica) [112,113], hollow polymeric nanosphere (HPN) [114], porous organic polymers [115] (covalent organic frameworks (COF) [116,117], hypercrosslinked polymers (HCP) [118], inherently microporous polymers (PIM) [119], porous aromatic frameworks (PAF) [120], conjugated microporous polymers (CMP) [121], and other biodegradable polymers (polypeptides) [122,123], MOF-based composites offer great opportunities for multifunctionalization by virtue of their unique organic–inorganic hybrid composite structures, ordered porous structures, and high biocompatibility [124]. When dealing with complex wounds, MOFs can achieve wound healing effects by releasing metal ions and loading drugs, including regulating the oxidative stress response, antibacterial effects, and promoting angiogenesis, collagen deposition, and re-epithelialization [125,126,127], by improving wound site infection and blood flow reperfusion to accelerate the wound healing process and restore normal skin function (Table 2).

### 6.1. MOF-Based Antibacterial Material

Delayed re-epithelialization exposes the wound to air for prolonged times and easily causes bacterial infections, with the most common bacterial species being coagulant-enzyme-negative Staphylococcus, Staphylococcus aureus, and Enterococcus [132]. The *G*^+^ bacteria *S. aureus* is the most common in diabetic ulcers [133]. A great deal of effort has been made to explore new antibacterial materials to overcome the dependence on antibiotics, for example, some antibacterial platforms based on nanomaterials, such as antibacterial nanoenzymes, nanofibers, and metal nanoparticles [42,134,135,136]. MOFs are mainly used as “reservoirs” for antimicrobial components to ensure the controlled release of metal ions, organic linkers, and loaded cargoes. 

The controlled methods mainly include four categories: (1) Since MOFs are formed through coordination between metals and organic linkers, certain types of metastable MOFs can easily collapse in liquids and then release metal ions and organic linkers to exert antibacterial effects. For example, copper-based MOFs (Cu-MOFs) are often unstable in solutions containing physiological proteins, hindering their direct use in wound healing. It has been proven that the toxicity of F-HKUST-1 modified with folic acid is significantly lower than that of HKUST-1, and the slow release of copper ions can be achieved [137]. (2) Photosensitive MOFs stimulate the release of MOF-loaded drugs through the photothermal effect by generating heat and free radicals under light irradiation. For example, the porphyrin-based MOF PCN-224 exhibits great absorbance in the wavelength range of 400–500 nm and can be activated to generate ^1^O_2_ to exert antibacterial effects [138]. The Zr-O clusters in UiO-66-NH_2_ can be activated to produce Zr^3+^ under UV light irradiation. When light is removed, Zr^3+^ gradually returns to Zr^4+^ by releasing electrons. Afterwards, the released electrons can react with the O_2_ and H_2_O adsorbed on the MOFs to generate ˙O^2−^ and H_2_O_2_ for an antibacterial effect [139]. (3) Some pH-sensitive MOFs can achieve controlled drug release in acidic infected areas, such as ZIF-8 MOFs, which are formed through the complexation between Zn^2+^ and 2-methylimidazole ligands. When ZIF-8 is deposited in an acidic environment, 2-methylimidazole is easily protonated, leading to the disruption of complex interactions and ultimately the collapse of the MOF structure, and at the same time, the loaded drug is rapidly released and begins to function. In a neutral environment, the drug is stably immobilized in the MOF nanoparticles and is not released [140]. (4) H_2_O_2_ triggers the release of drugs from MOF depots, such as MIL-100 (Fe), because H_2_O_2_ can be catalyzed by the Fe^3+^ metal center of MIL-100 (Fe), thereby dissociating the coordination bond between Fe^3+^ and the organic linker trimesic acid, resulting in the disintegration of the MIL-100 (Fe) structure [141]. MOF-based antibacterial material can overcome antibiotic resistance through physical contact, oxidative stress, the coordinated therapy of photothermal effects, and the antibacterial activities of metal ions or ligands. 

#### 6.1.1. Physical Contact

Silver sulfadiazine (AgSD) is widely used in topical antibiotic therapy to reduce the bacterial burden in burns and skin infections and promote wound healing because of its broad antibacterial spectrum [142,143,144]. AgSD releases Ag^+^ that binds to cellular components, leading to membrane damage and the disruption of DNA replication [145]. As a bacteriostatic agent, sulfadiazine (SD) inhibits the synthesis of folic acid by competing with bacteria for p-aminobenzoic acid [146]. The insolubility of AgSD limits its formulation with polymer carriers, resulting in reduced bioavailability and limited antibacterial activity [147]. In recent studies, the codelivery of ultrafine silver with soluble SD via a cyclodextrin metal–organic skeleton (CD-MOF) showed superior antibacterial activity to an insoluble AgSD [148]. UiO-66 was synthesized by green hydrothermal synthesis, and tetracycline (TC) was encapsulated by immersing it in an aqueous solution [149]. UiO-66 was integrated into polymer carboxymethyl cellulose (CMC) hydrogels to improve the mechanical properties and antibiotic release properties. CMC/TC@UiO-66 nanocomposite films were prepared by a simple casting method to achieve the controlled release of TC. Li et al. fabricated a novel composite nanofiber membrane with a uniform shape, good water resistance, and sustained drug release by electrostatic spinning. The membrane was loaded with β-CD COF enrofloxacin and flunixin meglumine, which both had antibacterial and anti-inflammatory properties and made wound healing easier [150]. In addition, the introduction of an MOF loaded with the antibacterial drug levofloxacin (LV) onto polyvinyl alcohol (PVA) membranes can be used to improve the water stability and antibacterial performance of nanofiber membranes (NFM) [151].

Nitric oxide (NO) is an intrinsic biological messenger that not only inhibits ubiquitination but also acts as an antibacterial agent [152]. However, its clinical application in the treatment of diabetic ulcers (DU) is hindered by the difficulty in controlling the release of NO. Yiqi Yang et al. proposed a smart near-infrared (NIR)-triggered NO nanogenerator (SNP@UCM) to precisely regulate the release of NO [153]. In this study, sodium nitroprusside (SNP) was used as the donor of NO, and upconversion nanoparticles (UCNPs) were used as the core of the nanogenerator to convert the deep-tissue-penetrating near-infrared signal into visible light in situ (Figure 6). The NO produced by SNP@UCM can react with the highly expressed ROS during wound healing to form ONOO^−^, thereby destroying the integrity of bacterial membranes. The use of microenvironmental ROS against *G*^−^ and *G*^+^ bacteria is an antibiotic-independent approach that will be investigated in future studies.

#### 6.1.2. Oxidative Stress

Skin ulcers are the most common cause of diabetes-related amputations, causing serious distress in the lives of patients [154]. Typically, these diabetic skin ulcers can be induced by chronic diabetic wounds, resulting in reactive oxygen species (ROS) overexpression and persistent inflammatory responses [155]. Because high blood sugar levels can cause bacteria to grow faster than in normal wounds and also prevent the immune system from killing invading bacteria, increased inflammation can lead to high oxidative stress, which further leads to a dramatic increase in reactive oxygen species, which triggers a chain reaction to destroy cells [3]. The disruption of the balance between ROS production and the antioxidant system is known as oxidative stress (OS) [156]. Many exogenous antioxidants can protect the body from the oxidative damage caused by excessive production of ROS, such as alpha-lipoic acid, curcumin, green tea, polyphenols, etc. [157,158]. For example, the study showed that α-lipoic acid may restore the endothelial cell function disrupted by hyperglycemia through the 3-MST/H_2_S pathway. A photothermal antibacterial platform composed of coated curcumin (Cur) nanocrystals can treat diabetic wounds infected with methicillin-resistant Staphylococcus aureus (MRSA) through an M2 (macrophage) Mφ polarization strategy [159]. In an acute kidney injury model, these green nanoparticle (tea polyphenol) scavengers can effectively prevent intracellular oxidative damage, accelerate wound healing, and protect kidneys from reactive oxygen species [160]. 

In recent years, natural antioxidants have been widely used in the anti-oxidative stress treatment of diabetes. Alpha-lipoic acid (α-LA), as a potent lipophilic antioxidant, has multiple functions, such as scavenging ROS, enhancing glucose uptake, and reducing apoptosis [157]. To increase the absorption and bioavailability of α-LA and improve its therapeutic effect, Li et al. developed a QCS-OHA-α-LA hydrogel using quaternary ammonium chitosan (QCS) and oxidative hyaluronic acid (OHA) hydrogel acting as a scaffold, incorporating K-γ-CD-MOFs to deliver α-LA to the wound edge for sustained release, thereby maximizing the effect of α-LA [136]. Among them, γ-cyclodextrin (γ-CD) has a hydrophobic cavity, which is beneficial for the loading and sustained release of hydrophobic drugs such as lipoic acid. Studies have shown that curcumin can scavenge ROS in activated macrophages under diabetic conditions [161,162], possibly by inhibiting the expression of related neuropeptides (Cgrp, Sst, and Sp) and chemokines (Ccl2 and Ccl3) in DRG and regulating the PI3K/AKT and MAPKs signaling pathway to relieve the pain hypersensitivity reaction caused by diabetes peripheral neuropathy (DPN) and protect against peripheral nerve damage [163]. Coating curcumin into γ-cyclodextrin metal–organic frameworks (γ-CD-MOFs), along with the dissociation of CD-MOFs, increases the water solubility of curcumin due to the inclusion of CD [164,165]. Since tea polyphenols (TP) are soluble in water, they are easily oxidized. Therefore, Chen et al. encapsulated TP in a PVA/alginate hydrogel (TPN@H) and demonstrated that TPN@H could promote wound healing in diabetic rats by regulating the PI3K/AKT signaling pathway [166]. Peng et al. developed a green nanoparticle scavenger against oxidative stress based on TP, which effectively prevented intracellular oxidative damage and accelerated wound recovery [160]. A recent study utilized C-confined CoOx NPs to mimic peroxidase enzymatic activity to scavenge hydroxyl radicals [167]. Although the C-confined CoOx NP nanozyme exhibited peroxidase-like activity, the enzymatic activity was still lower than that of natural horseradish peroxidase (HRP), which needs to be further improved.

#### 6.1.3. Photothermal Effect

Recently, a photothermal therapy (PTT) based on near-infrared (NIR) light irradiation has been widely used for bacterial infections, especially multidrug-resistant bacterial infections [168,169,170,171]. A bacterial protein denaturation induced by locally elevated temperature (over 50 °C) for the introduced photothermal agents will cause irreversible bacterial structural damage and bacterial death [172]. Compared with current antibiotic therapy, PTT has broad-spectrum bactericidal efficiency through physical heating, with rapid sterilization and minimal damage to healthy tissue [173]. However, when the temperature increase caused by PTT fails to completely eliminate bacteria, PTT alone may require higher temperatures to completely eliminate bacterial populations, which may damage adjacent healthy tissue [174]. At the same time, lower temperature activation is not enough to make it difficult to achieve significant photothermal ablation of bacteria.

MOFs are a class of stable crystalline porous coordination compounds formed by the assembly of metal ions and organic ligands in a modular and simple manner [175]. Although it is easy to introduce a variety of antibacterial metal ions into MOFs for chemical bacterial ablation, this single-mode sterilization method still suffers from the problems of large dosage, low antibacterial efficiency, and slow sterilization [176]. MOFs are able to combine the inherent physical and chemical properties of inorganic and organic photonic units, allowing the realization of good photonic functional applications [175,177,178,179]. Under the excitation of near-infrared light, the structure of MOFs facilitates the rapid transfer and separation of photogenerated electrons and holes and promotes the efficient utilization of electrons [180]. In addition, the pores inside MOFs can also act as photonic units to encapsulate a large number of guest species, which is beneficial to fully understanding the mechanism of the energy transfer process within the framework [16]. The vast combinatorial possibilities, synergistic effects, and controllable ordering of multiple photonic units (MPUs) differentiate them from other inorganic and organic photonic materials as a promising platform for novel photonic functional applications [179]. That is, the structural tunability of photonic MOFs through various doping methods can be used to control their light-harvesting capabilities. Therefore, it is necessary to develop multimodal therapies to take advantage of PTT and MOFs.

Combining PTT and MOFs for antibacterial therapy has become a research hotspot in recent years. Prussian blue (PB) and its analogs are an important class of MOFs. Li et al. developed an exogenous antibacterial agent composed of zinc-doped Prussian blue (ZnPB) that optimized the synergistic antibacterial effect of ZnPB for photothermal therapy and ion release [16]. Furthermore, ZnPB treatment resulted in the upregulation of genes involved in tissue remodeling, promoting collagen deposition and enhancing wound repair. Xiao et al. synthesized a NIR/pH dual stimulus-responsive platform (Van@ZIF-8@PDA) using polydopamine (PDA) as a photothermal agent to encapsulate vancomycin in the pores of ZIF-8 for a synergistic photothermal/drug antimicrobial therapy of zeolitic imidazolate frameworks [181]. This method takes full advantage of MOF and PDA. Zhang et al. synthesized hyaluronic-acid-grafted dopamine (HA-DA) by amidation reaction, loaded a nanoscale MOF photosensitizer zirconium-based porphyrin porous coordination network (PCN-224) on the surface of black phosphorus (BP) nanosheets, and synthesized a photoinitiated physical/chemical double-crosslinked injectable hydrogel (HA-DA/Fe^3+^/PCN@BP) [182]. Under 660 nm laser irradiation, it can be effectively sterilized using a relatively small amount of ROS under 50 °C (Figure 7A).

A large number of studies have proven that the therapeutic effect of PTT under mild conditions is limited [183]. Therefore, it is necessary to find new strategies to improve the therapeutic effect of PTT. To address this issue, Wang et al. decided to use amoxicillin (AMO), a beta-lactam antibiotic, to disrupt bacterial cell walls first, followed by PTT [184] (Figure 7B). First, a Pd-Cu nanoalloy (PC) and AMO were intercalated into pH-responsive ZIF-8 in situ synthesis. AMO molecules can be embedded during the assembly process. Under acidic conditions, the ZIF-8 backbone was continuously degraded, and AMO was released from it. Due to its good photothermal properties, PC can effectively absorb NIR and convert it into heat to kill bacteria and destroy the integrity of bacterial biofilms. Combining Zn-doped MoS_2_ nanosheets with ZIF-8 MOFs, an environmentally friendly hybrid material with enhanced photocatalytic performance and photothermal effect can be designed to rapidly and effectively kill bacteria while accelerating wound healing [185]. Pseudoperoxidase (POD) nanozymes can disrupt the integrity of cell membranes and lead to bacterial death by regulating intracellular ROS levels [186]. The synergistic combination of the enzymatic and photothermal properties of nanozymes makes resistant bacteria more vulnerable, makes wounds heal more easily, and reduces the risk of inflammation [187]. Although this synergistic effect exhibits more excellent bactericidal activity, it is still difficult to break through the requirement of POD mimicking the strongly acidic conditions (pH 3.0–4.0) catalyzed by nanozymes [180] (Figure 7C). The introduction of MOFs seems to be effective in improving the strongly acidic conditions [180,188]. To prove this theory, Liao constructed Zr-MOF-based UiO-66-NH-CO-MoS_2_ nanocomposites (UNMS NCs) via amide bonds and proposed a POD-like mimetic enzyme based on photothermal and photodynamic modulation with mechanisms against bacteria under a single NIR irradiation (808 nm). Both the photodynamic properties and nanoenzyme activity of UNMS NCs can lead to the generation of reactive oxygen species. UNMS NCs showed high catalytic activity in the pH range of 4.0−7.0 and exhibited excellent antibacterial activity against ampicillin-resistant *E. coli* and methicillin-resistant *S. aureus*. It is a pleasant surprise that the high temperature induced by 808 nm laser radiation can accelerate the oxidation process of glutathione, which is more likely to lead to bacterial death. Numerous studies have shown this platform as a potential alternative to current antibiotic therapy for bacterial wound infections.

#### 6.1.4. Metal Ions or Ligands 

Antibiotics are important antibacterial drugs and have been widely used in the treatment of bacterial infections. However, existing antibiotics have high resistance rates to single and multiple drugs [189], leading to “superbugs” that threaten human health. Therefore, it is urgent to develop antibacterial materials with inherent antibacterial properties and low toxicity without drug resistance to solve the problem of wound site infection. Over the past decade, metallic antibacterial agents such as zinc-, silver-, and copper-based oxides or ions have been used to kill or inhibit bacterial growth on wounds [151,176,190] (Table 3).

### 6.2. MOF-Based Provascular Materials

A high-glucose environment can damage microvascular endothelial cells and induce vasodilation dysfunction and tissue ischemia, resulting in slow wound healing [108,199]. Up to 25% of people with diabetes are at risk of chronic nonhealing wounds, such as diabetic foot ulcers and pressure ulcers, and at least 68% of people with diabetic wounds will die from systemic infections within 5 years [200]. Systemic administration may not deliver adequate drug delivery to target tissues due to insufficient perfusion caused by insufficient wound angiogenesis. Therefore, promoting angiogenesis and inhibiting bacterial infection are crucial for the treatment of chronic wounds in diabetes.

#### 6.2.1. Drugs Act in Coordination with Metal Ions

To accelerate skin regeneration, Wang et al. used the natural biomolecules of niacin (NA) as organic ligands for the first time to prepare a novel copper-based MOF material (CuNA) with a ring structure for loading and releasing bFGF, which significantly promoted cell proliferation and angiogenesis [201]. As one of the essential metal elements in the human body, the improved effect of copper-based MOFs on diabetic wound healing has been demonstrated in previous reports [201,202,203]. Copper promotes the release of vascular endothelial growth factors (VEGFs) and cytokines by coordinating the expression of hypoxia-inducible factor-1a (HIF-1a) and proline hydroxylase, triggering endothelial cell angiogenesis [204,205]. Copper ions can also stimulate the expression of several important factors, including MMP-1 and IL-8, promoting wound healing and skin aging by inducing the expression of MMP-1 [202], while at higher concentrations, copper enhances the expression of IL-8 and limits the action of MMPs, which may favor fibroblast proliferation and extracellular matrix deposition [206]. In addition, copper can also act as an antibacterial agent [137], promoting wound healing by reducing infection. However, the rapid release of copper ions is prone to cytotoxicity, and how to stabilize and slow the release of copper ions has quickly become the focus of attention. Cu-MOFs constructed from ions and biomolecules can solve the problems of the sustained release of copper ions and the biocompatibility of organic ligands. For example, Xiao et al. stabilized HKUST-1 by modifying HKUST-1 with folic acid, increasing the hydrophobicity of MOF and reducing the BET surface area, resulting in a slow release of Cu^2+^ and reducing the toxicity of HKUST-1 in vitro and in vivo [207]. A novel antioxidant HKUST-1 heat-responsive polyglycol citrate hydrogel composite that stabilizes HKUST-1 NPs in a protein solution was shown to promote wound healing in diabetic mice [203]. Among them, poly(polyethyleneglycol citrate-co-N-isopropylacrylamide) (PPCN) is a liquid at room temperature (22 °C) and has a lower critical solution temperature (LCST) between 26–28 °C, depending on the medium. The PPCN-HKUST-1 interaction enables the sustained release of copper ions and maintains the thermal response and antioxidant properties, and the use of HKUST-1 does not affect the gelation properties of PPCN (Figure 8).

#### 6.2.2. NO

It is known that NO is produced by organisms, including bacteria, plants, and animals [172]. Nitric oxide (NO) is an intrinsic biological messenger that not only acts as an antibacterial agent [152] but also inhibits ubiquitination, promotes angiogenesis, and stimulates collagen deposition [41,208,209]. In mammals, NO is synthesized from L-arginine, oxygen, and NADPH by various NO synthases (NOS) [210]. A series of studies have shown that hyperglycemia in diabetic patients can inhibit the synthesis of endogenous NO, which is one of the main reasons for the slow healing of diabetic wounds [211,212]. Existing studies have summarized a variety of NO donors for NO storage and transport, including organic nitrates and nitrites, metal-NO complexes, diazenium diols (NONOates), and S-nitrogenates [213,214]. Currently, NO is used as a gas drug for the treatment of diabetic wounds [213,215,216], but its clinical application in the treatment of diabetic ulcers (DU) has been hindered by the difficulty in controlling the release of NO. To solve this problem, Yiqi Yang et al. proposed a smart near-infrared (NIR)-triggered NO nanogenerator (SNP@UCM) to precisely regulate the release of NO [153]. In this study, sodium nitroprusside (SNP) was used as the NO donor, and upconversion nanoparticles (UCNPs) were used as the core of the nanogenerator to convert the deep-tissue-penetrating near-infrared signal into visible light in situ (Figure 9). SNP@UCM inhibits the ubiquitination-mediated proteasomal degradation of HIF-1α by inhibiting its interaction with E3 ubiquitin ligase under NIR irradiation, increases HIF-1α expression in endothelial cells, enhances angiogenesis at wound sites, and promotes vascular endothelial growth factor (VEGF) secretion, cell proliferation, and migration. Based on the basic idea of NIR for controllable release, Shun et al. used responsive graphene oxide (GO) to encapsulate HKUST-1, obtained NO@HKUST-1@GO microparticles (NHG), and loaded them into typical release platforms. A controllable NO release platform has also been successfully prepared in the porous MN patch based on polyethylene glycol diacrylate (PEGDA) [208]. In addition, Zhang et al. introduced a copper-based MOF, namely HKUST-1, as a NO-loading carrier, and designed a sustained-release system of NO with a core–shell structure using the electrospinning method [213]. The additional copper ions released by degradable HKUST-1 synergized with NO to promote endothelial cell growth and significantly improve the angiogenesis, collagen deposition, and anti-inflammatory properties of the wound bed, ultimately accelerating the healing of diabetic wounds.

## 7. Conclusions and Future Perspectives

MOF-based therapeutic platforms have been extensively studied in diabetes and diabetes-complicated chronic wounds, but few have been tested in the clinic. There are still problems, such as unclear therapeutic targets, a lack of systematic toxicity detection and analysis, and difficulty in comprehensively dealing with complex wounds. In this review, different synthesis methods for MOFs are briefly introduced, highlighting the advantages and disadvantages of each method. Based on the application of MOFs in the biomedical field, the development of different types of sensors in the field of diabetes diagnosis and the potential value of the glycemic responsive regulation of MOFs are mainly updated. We also summarize the latest MOFs as delivery platforms for the repair of complex wounds, including different antibacterial and provascular strategies. Finally, the potential toxicity of MOFs is also an issue that we cannot ignore. We believe that this review has important reference value for the future development of smart, nontoxic, easily accessible, and low-cost MOF-based therapeutic platforms.

Due to their outstanding chemical and physical properties, MOFs have been the focus of extensive research for various applications. Despite the special properties of MOFs that facilitate their use as platforms for disease diagnosis and drug delivery, several challenges remain in this field. Research reports on the biological applications of these materials are limited. Toxicity, targetability, degradability, blood half-life, stability, and potential toxicity are all issues that we cannot ignore.

The toxic effects of MOFs may be due to the presence of metal ions and ligands in their own composition [217]. The toxicity of metal ions such as Ag, Cu, Au, Zn, and Fe in MOFs designed for drug delivery or other therapeutic applications is usually dose-dependent. Heavy metals such as lead, arsenic, chromium, and cadmium as well as toxic carboxylates, phosphonates, phenates, sulfonates, and amines are all common ligands contained in MOFs. They will cause certain negative effects when they enter body for metabolism and degradation [218]. In addition to these factors, the crystal size of the MOF and the solvent used in synthesis may also contribute to toxicity. The modification of MOFs may not be limited to the use of different metals or ligands, as the size/shape of MOF particles themselves can also be used as a tool to achieve new properties. Reducing the particle size of MOFs results in a significant increase in their external surface area can significantly improve the surface-area-to-volume ratio as well as their chemical reactivity [219]. Moreover, the properties of MOF nanoparticles are markedly different from those of their bulk analogs with the same chemical composition. Thus, the toxicity of a material may be amplified by a size reduction due to its ability to cross physiological barriers, altering its biodistribution and elimination, making it much more dangerous than its bulk analogs [220]. These potential toxicities will affect whether MOFs can be successfully applied in clinical applications. Therefore, we must consider and solve the following problems in the future work:

1. In addition to direct therapeutic delivery, we must consider whether the surface of an MOF can be further modified to better achieve biological targeting and improve its efficiency in the process of disease diagnosis and treatment.

2. Taking into account the biodegradability of synthetic metals and ligands, tracking cellular uptake, in vivo metabolism, and excretion, we must consider the reasonable design of the shape, type, and the number of functional groups of an MOF to improve application performance.

3. We fabricate MOFs with particle sizes less than 200 nm so that these drug carriers can circulate freely within the smallest capillaries [220]. At the same time, the drug loading rate of MOFs was improved. In the process, however, continuously reducing the synthetic size amplifies their physical and chemical reactivity, increasing the risk of potential toxicity. Therefore, we must analyze the biocompatibility of nanoscale MOFs in the in vivo environmental state and not be fooled by other excellent properties.

MOFs will be one of the most promising materials in the biomedical field in the future, and it is hoped that this work will pave the way for more advanced research in this field.

## Figures and Tables

**Figure 1 biomolecules-12-01240-f001:**
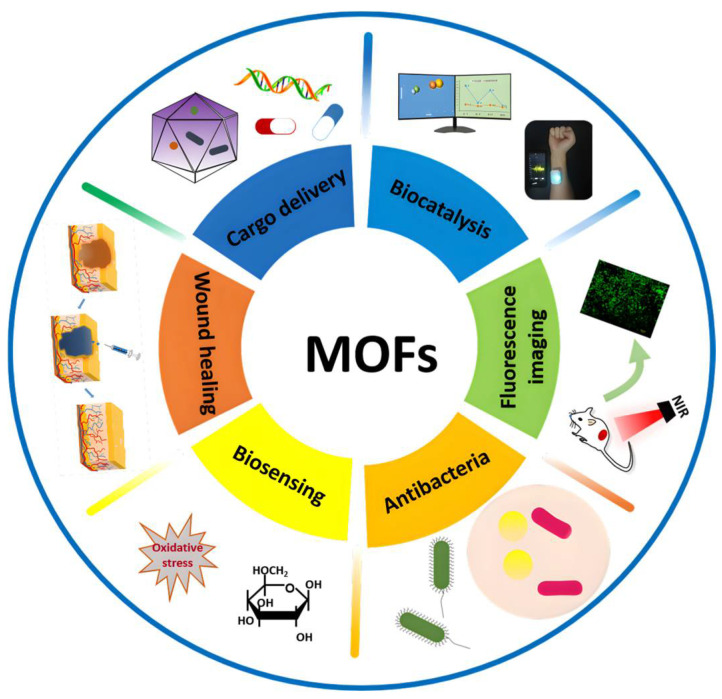
Applications of MOFs in the biomedical field.

**Figure 2 biomolecules-12-01240-f002:**
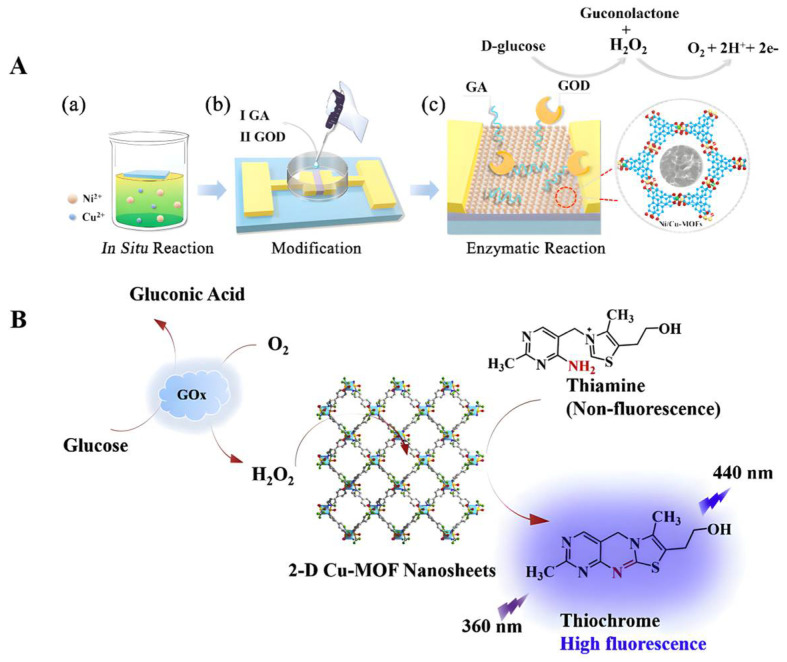
(**A**) Schematic diagram of the fabrication, modification, and principle of the glucose detection of GOD-GA-Ni/Cu-MOFs-FET. (**a**) FET was suspended on the surface of a reaction solution with mixed metal ions. Insert shows the response time of the sensor to glucose. (**b**) The Ni/Cu-MOFs were modified by dropping GA and GOD sequentially for cultivation. (**c**) Ions generated from the enzymatic reaction of glucose accumulated on the surface of bimetallic MOF films, inducing a change in charge concentration. Reproduced with permission [67]. Copyright 2021, Journals & Books. (**B**) The working mechanism of 2D Cu-MOF nanosheets. Reproduced with permission [75]. Copyright 2019, *Analytica Chimica Acta*.

**Figure 3 biomolecules-12-01240-f003:**
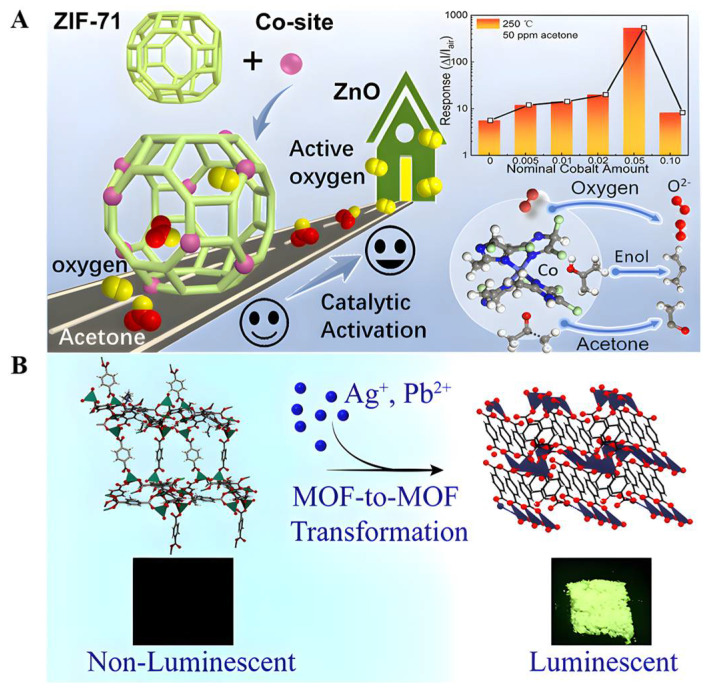
(**A**) Schematic illustration of the catalytic gas-sensing mechanism in ZnO@ZIF-71(Co) with active Co sites. Reproduced with permission [86]. Copyright 2020, *ACS applied materials & interfaces*. (**B**) Facile and fast transformation of nonluminescent to highly luminescent metal–organic frameworks. Reproduced with permission [56]. Copyright 2021, *ACS Applied Materials & Interfaces*.

**Figure 4 biomolecules-12-01240-f004:**
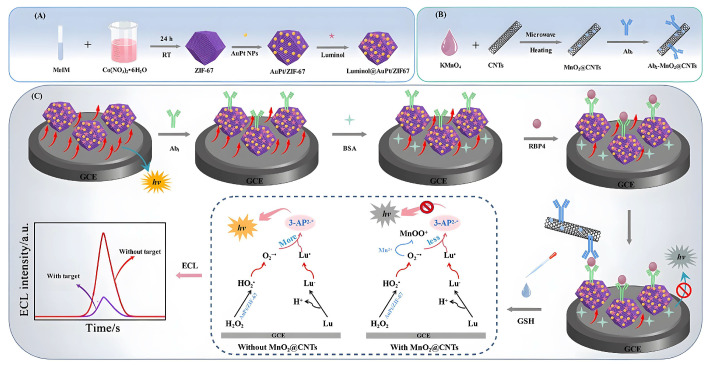
Principle of the dual-quenched ECL immunosensor based on luminol@AuPt/ZIF-67 and MnO2@CNTs for the detection of RBP4. (**A**) The preparation procedure of the luminol@AuPt/ZIF-67. (**B**) The synthesis route of Ab_2_-MnO_2_@CNTs. (**C**) The detection process and operation mechanism of the sandwich-type immunosensor. Reproduced with permission [59]. Copyright 2021, *Journal of Nanobiotechnology*.

**Figure 5 biomolecules-12-01240-f005:**
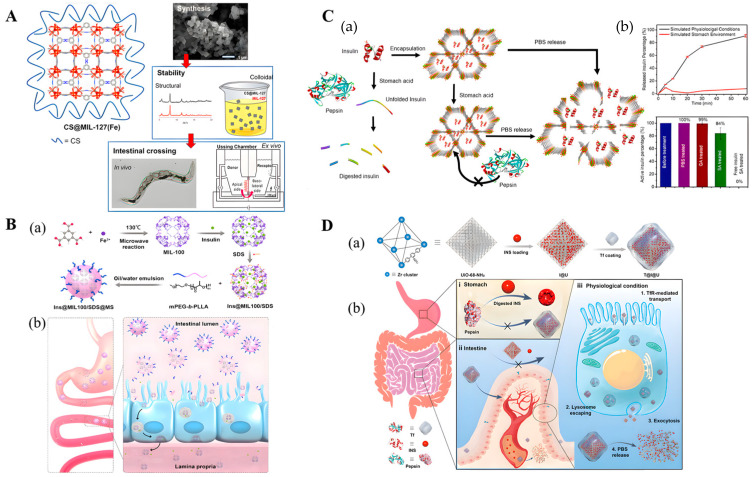
(**A**) **Left**: Schematic view of the structure of CS@MIL-127 nanoparticles (NPs). **Right**: Procedure for intestinal crossing evaluation: (**top**) synthesis of the NPs; (**middle**) evaluation of the structural, chemical, and colloidal stability under simulated oral conditions, depicting an example of the structural stability in mucin-complemented simulated intestinal fluid (lis-SIF-muc); and (**bottom**) direct observation of the NP bypass in the C. elegans model and scheme of a Ussing chamber used in the ex vivo experiments with the intestine of rat. Reproduced with permission [97]. Copyright 2022, *ACS Nano*. (**B**) (**a**) Schematic illustration of the preparation process of nanocomposite Ins@MIL100/SDS@MS. (**b**) Schematic illustration of the transit through the stomach of Ins@MIL100/SDS@MS and the subsequent dissolution of the microspheres in the intestine, rendering exposure and penetration through the intestinal epithelium of Ins@MIL100/SDS. Reproduced with permission [101]. Copyright 2020, *ACS Applied Materials & Interfaces*. (**C**) Schematic representation of (**a**) encapsulation of insulin in the mesopores of NU-1000 and exclusion of pepsin from the MOF framework. (**b**) Exposure of free insulin and insulin@NU-1000 to stomach acid. Free insulin denatures in stomach acid and is digested by pepsin. Insulin@NU-1000 releases insulin when exposed to a PBS solution. Reproduced with permission [102]. Copyright 2018, *Journal of the American Chemical Society*. (**D**) Schematic representation of Tf-coated acid-resistant nMOF nanosystem for oral delivery of INS. (**a**) The synthesis of the UiO-68-NH_2_-based nanosystems. (**b**) The oral delivery process of the Tf-coated UiO-68-NH_2_ nanosystem in vivo addresses both the harsh environment in the stomach (**i**) and the epithelial cell layer barriers (**ii**). (**iii**) The Tf-coated UiO-68-NH_2_ nanosystem contributed to overall intensive intestinal cell absorption under physiological conditions. Reproduced with permission [103]. Copyright 2022, *Science advances*.

**Figure 6 biomolecules-12-01240-f006:**
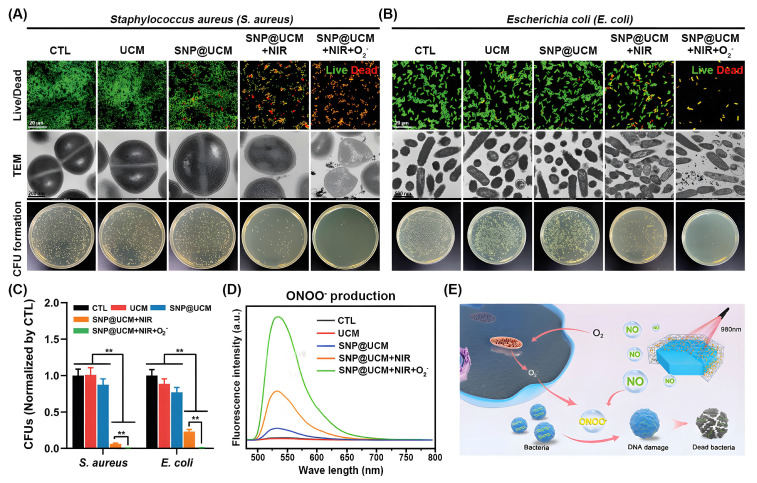
In vitro antibacterial activity of SNP@UCM nanogenerators. (**A**,**B**) Antibacterial activity of the indicated treatments against *S. aureus* (**A**) and *E. coli* (**B**) as determined by bacterial live/dead staining, transmission electron microscopy (TEM) analysis, and colony-forming unit (CFU) counts. (**C**) Quantitative analysis of CFUs. (**D**) ONOO^−^ production by bacteria under the indicated treatments. (**E**) Schematic of the proposed antibacterial mechanism. NO released from SNP@UCM reacted with ROS produced during the bacterial invasion to form ONOO^−^. ONOO^−^ destroyed bacterial membrane integrity. ** *p* < 0.01. Reproduced with permission [153]. Copyright 2021, *Advanced materials*.

**Figure 7 biomolecules-12-01240-f007:**
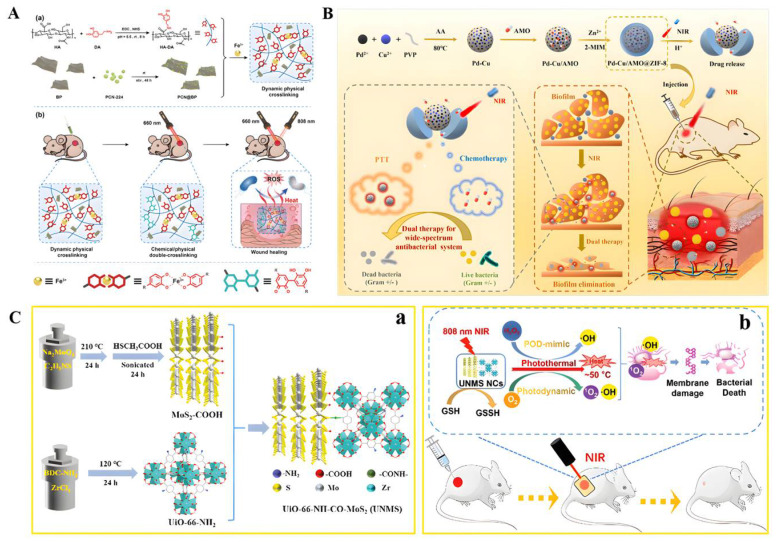
(**A**) Schematic illustration of the injectable hydrogel with tunable mechanical properties and a physical/chemical double network that promotes wound healing while combatting bacterial infections. (**a**) The synthesis process for the HA-DA/Fe^3+^/PCN@BP dynamic physical cross-linking injectable hydrogel through catechol-Fe^3+^ coordination cross-linking. (**b**) The HA-DA/Fe^3+^/PCN@BP injectable hydrogel introduces chemical cross-linking in situ under laser irradiation, enabling the regulation of its mechanical properties and combining PDT/PTT synergistic antibacterial actions to promote wound healing. Reproduced with permission [182]. Copyright 2022, *Advanced Healthcare Materials*. (**B**) Schematic diagram of the dual stimuli-responsive chemo-photothermal combination system based on PC for the procedural antibacterial therapy. Reproduced with permission [184]. Copyright 2022, *Acta Biomaterialia*. (**C**) (**a**) Detailed preparation of UNMS NCs; (**b**) Schematic illustration of the bactericidal mechanism of UNMS NCs and the treatment of wound infection. Reproduced with permission [180]. Copyright 2021, *Advanced Healthcare Materials*.

**Figure 8 biomolecules-12-01240-f008:**
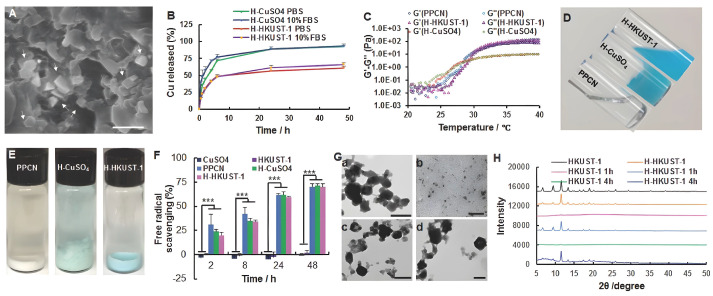
Characterization of H-HKUST-1. (**A**) SEM digital image of H-HKUST-1. White arrows point to HKUST-1 NPs. (Scale bar: 500 nm). (**B**) Copper release from H-HKUST-1 and H-CuSO_4_ in PBS or 10% FBS. (**C**) Rheological characterization of PPCN, H-HKUST-1, and H-CuSO_4_. The storage modulus G′ and loss modulus G″ were plotted logarithmically against temperature (20–40 °C at 10 Hz) for the corresponding hydrogel samples. (**D**) Digital images of PPCN, H-CuSO_4_, and H-HKUST-1 at 22 °C. (**E**) Digital images of PPCN, H-CuSO_4_, and H-HKUST-1 after incubation in 10% FBS at 37 °C. (**F**) ABTS radical scavenging capacity of CuSO_4_, HKUST-1 NPs, PPCN, H-CuSO_4_, and H-HKUST-1 (n = 3, *** *p* < 0.001). (**G**) TEM showing the morphology of (**a**,**b**) HKUST-1 NPs and (**c**,**d**) H-HKUST-1 before (**a**,**c**) and after (**b**,**d**) incubation in 10% FBS at 37 °C. (**H**) XRD patterns of HKUST-1 NPs and H-HKUST-1 before and after incubation in 10% FBS. Reproduced with permission [203]. Copyright 2017, *Advanced Functional Materials*.

**Figure 9 biomolecules-12-01240-f009:**
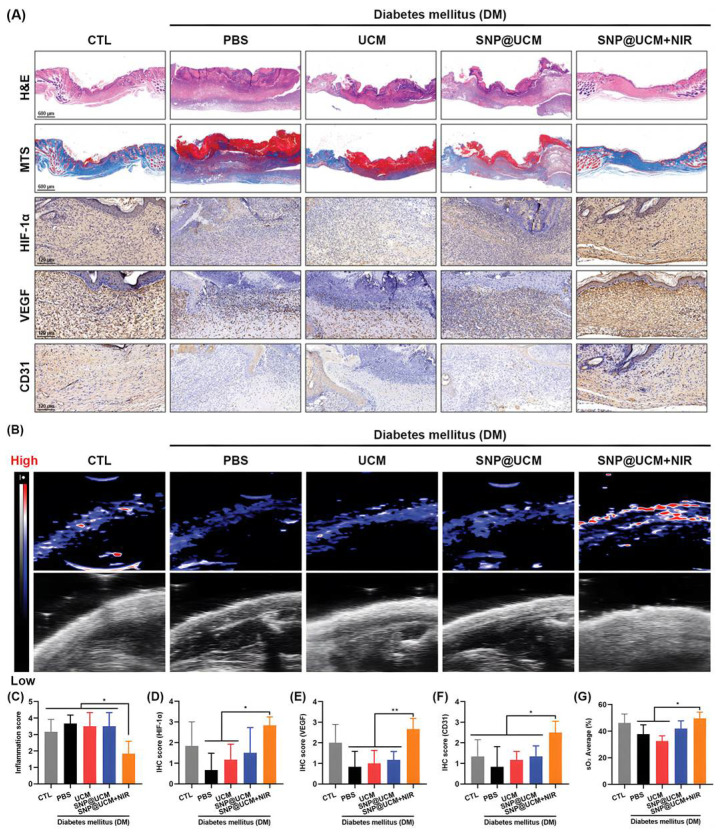
Accelerated in vivo angiogenesis and wound healing of infected diabetic ulcers in response to SNP@UCM treatment. (**A**) Hematoxylin and eosin (H&E) staining, Masson’s trichome staining, and immunohistochemistry (IHC) of wound tissues on day 14. (**B**) Ultrasound and photoacoustic images of wound tissues on day 14. (**C**) Inflammation score based on H&E staining. (**D**–**F**) Semi-quantitative analysis of IHC. (**G**) Semi-quantitative analysis of photoacoustic images. sO2: oxygen saturation. * *p* < 0.05, ** *p* < 0.01.Reproduced with permission [153]. Copyright 2021, *Advanced materials*.

**Table 1 biomolecules-12-01240-t001:** Comparison of advantages and disadvantages of MOF synthesis methods.

Synthetic Methods	Advantages	Disadvantages	Ref.
Hydrothermal/solvothermal synthesis	Generality, simplicity, and low cost. Can large-scale preparation at mild temperatures.	Long reaction time, high temperature requirement	[22,27,28]
Microwave-assisted synthesis	Shorten the reaction time to a few hours or even several minutes. No excessive by-products and the high purity and small size of MOFs obtained.	Reaction solvent requirements are limited	[29,30,31]
Room-temperature synthesis	Reaction conditions are simple and can be prepared on a large scale.	Limited range of adaptation	[32,33]
Ultrasonic-assisted synthesis	Quickly disperse solutes and speed up the reaction process, improve the reaction efficiency.	Hard to obtain a lot of product	[34,35]
Mechanochemical synthesis	Safety, environmental protection	Hard to obtain a lot of product	[36,37]
Microfluidic synthesis	The effective mixing of organic phase and inorganic phase is realized.	Limited range of adaptation	[38,39]
Biomimetic mineralization	Beneficial to enhance the robustness and biocompatibility of biomacromolecules.	Only suitable for biomacromolecules that are similar in size to the pore size of MOFs and are solvent resistant in the preparation of MOFs.	[25,26,40]

**Table 2 biomolecules-12-01240-t002:** Comparison of different porous materials or biomaterials.

Porous Materials	Composition	Advantages	Disadvantages	Ref.
Metal–organic framework (MOF)	Organic ligands and their coordinated metal ions/ion clusters	Ordered porous structure, biocompatibility, and ease of functional modification.	Targeting and potential biotoxicity.	[124,128]
Mesoporous silica nanoparticles (MSN)	Silica (SiO_2_)	Huge loading capacity, controllable particle size and shape, suitability for easy functionalization, and biocompatibility.	Poor dispersibility and stability, prone to accumulation, and requires modification. A fully reversible lid is required to close the pore access.	[113]
Hollow polymeric nanosphere (HPN)	1,4-Bisbenzenedimethanol (BDM), 1,2-dichloroethane (DCE)	Has uniform hollow spherical shape, sufficient surface area, and excellent physicochemical stability.	The synthesis scale is small, the cost is high, the controllability is poor, and the mechanism research is not in-depth.	[114]
Poly (α-L-glutamic acid) (PGA)	L-glutamic acid	Inherent biodegradability, biocompatibility, and ion charging characteristics.	The synthesis and purification procedures are complex, and the scalability and repeatability need to be improved.	[123]
Poly (L-histidine) (PLH)	α-Amino acid N-carboxylic anhydride (NCAs)	Positive charge, suitable for PH-triggered targeted drug delivery.	The synthesis process has poor repeatability and multiple molecular weight distributions. Unpredictable drug coupling sites increase the heterogeneity of PLH.	[122]
Covalent organic frameworks (COFs)	Light elements (H, C, N, O, B)	Large surface area, high thermal stability, good biocompatibility, and good biodegradability.	The synthesis condition is not mild enough, the preparation cost is high, and the structure is uncontrollable.	[116]
Hyper-crosslinked polymers (HCPs)	Light elements (H, C, N, O, B)	High specific surface area, mild synthesis conditions, a wide range of monomer sources, cheap and easy to obtain catalyst.	Modification strategies and synthesis methods need to be improved.	[118]
Polymers of intrinsic microporosity (PIMs)	Light elements (H, C, N, O, B)	Highly microporous	The resulting materials are amorphous and have a wide pore size distribution, which is not easy to adjust and control.	[119,129]
Porous aromatic frameworks (PAFs)	Light elements (H, C, N, O, B)	High stability, large specific surface area, large pore volume, strong modifiability.	Locally ordered and long-range disordered skeleton structure	[120,130]
Conjugated microporous polymers (CMPs)	Light elements (H, C, N, O, B)	Multi-micropore, high surface area, chemical stability, thermal stability, structure adjustable.	Expensive production costs	[131]

**Table 3 biomolecules-12-01240-t003:** MOFs in antibacterial treatments.

MOF	MOF Skeleton Components	Antibacterial Composition	Pathogenic Bacteria	References
MIL-53	Fe^3+^, terephthalic acid, chitosan	Vancomycin	*S. aureus*	[10]
SNP@UCM	SNP, ssPDA, UCNP	NO, ROS	*S. aureus* and *E. coli*	[153]
Cu-MOFs	Cu^2+^, ribose, chloramphenicol	CHL, Cu^2+^	*E. coli* and *P. aeruginosa*	[191]
Zn-MOF	Zn^2+^, lactobionic acid	Amoxicillin, Zn^2+^	*H. pylori*	[192]
nFMs@Amp	Fe^3+^, pluronic F-127	•OH	*S. pneumoniae*	[193]
PCN-224 MOFs	Zr^4+^, pullulan, polyvinyl alcohol	^1^O_2_	*E. coli* and *S. aureus*	[194]
LV@UiO-66-NH_2_@PVA	Nanofibrous membranes, UiO-66-NH_2_, polyvinyl alcohol	Levofloxacin	*E. coli* and *S. aureus*	[151]
FSZ-Ag	Ag^+^, Zn^2+^, 2-methylimidazole	Ag^+^, Zn^2+^	*E. coli* and *S. aureus*	[195]
C-Zn/Ag	Ag^+^, Zn^2+^, 2-methylimidazole	Ag^+^, Zn^2+^	*E. coli* and *S. aureus*	[176]
Ag-Phy@ZIF-8@HA	Ag^+^, Zn^2+^, Physcion, 2-methylimidazole	Ag^+^, Physcion	*E. coli* and *S. aureus*	[196]
BMOF-DMR	Cu^2+^, Zn^2+^, 2-methylimidazole	Cu^2+^, Zn^2+^	*E. coli* and *S. aureus*	[197]
PCN@BP	Zr^4+^, TCPP, benzoic acid, DMF, BP	ROS	*E. coli* and *S. aureus*	[182]
CaO_2_/GQDs@ZIF-67	Co^2+^, 2-methylimidazole, CaO_2_	•OH	*E. coli* and *S. aureus*	[198]
Au^3+^-UiO-67	Au^3+^, Zr^4+,^ 2,2′-bipyridine-5,5′-dicarboxy acid	•OH, ^1^O_2_	*E. coli* and *S. aureus*	[126]

## Data Availability

Not applicable.

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
