# Peer review of "Application of Metal–Organic Framework in Diagnosis and Treatment of Diabetes"

_biomolecules, 2022, doi:10.3390/biom12091240_

Round 1

Reviewer 1 Report

This article reviews application of metal-organic framework in diagnosis and treatment of diabetes and its wound healing. In this review, the authors first introduced the different synthesis methods for MOFs, highlighting the advantages and disadvantages of each method. And then, the applications of MOFs in the biomedical field were presented, the development of different types of sensors in the field of diabetes diagnosis and the potential value of MOFs in glucose response regulation were analyzed. In particular, it summarizes the latest MOF as a delivery platform for repairing complex wounds, including different antimicrobial and vascular approaches. Finally, the potential toxicity of MOF has not been solved, and the future prospects and corresponding research directions are given. In general, the logic of the article is rigorous, and the content is relatively complete.

1. The picture of the whole article is small and unclear, the font in the picture is blurred, and the font size should be one size smaller.

2. What are the advantages or disadvantages of MOF compared with other porous materials or biomaterials in the specific process of wound repair? Give specific examples and discuss them? For example, silica materials, porous organic polymers, and other biodegradable polymers such as polypeptides.

3. Some related research about the biomedical materials should be cited and compared with them. Advanced functional materials, 2020, 30(2): 1902634. Nano Res. 15, 5556–5568 (2022). https://doi.org/10.1007/s12274-022-4160-6. Biomolecules, 2022, 12(5): 636. Biomater. Sci., 2022, DOI: 10.1039/D2BM00719C.  J Control Release. 2022 Aug 6:S0168-3659(22)00485-0. doi: 10.1016/j.jconrel.2022.08.005. Journal of the American Chemical Society 140.42 (2018): 13534-13537.

4. When MOF is used as a "reservoir" for antimicrobial components, how does it specifically control the stability and dissolution of its structure without affecting the guest molecules?

Reviewer 2 Report

The article summarizes the synthesis of MOFs and application of MOFs in Diabetes Diagnosis, regulating blood glucose, and wound healing. And the title of this paper is “Application of metal-organic framework in diagnosis and treatment of diabetes and its wound healing”. The title and content are not appropriate enough and should be adjusted.
(1) In this review, the author only listed the synthesis methods of some MOFs, and listed some advantages and disadvantages of different synthesis methods. There have been many corresponding reviews. The author should review the recent development of some MOFs, and this part should not include the scope of the review topics.
(2) In the paper, the logic of many narratives is problematic, such as lines 131-132 the author describes a large number of MOFs based immunosensors however, there are only two references.
(3) For the abbreviations of metal-organic framework materials, there are many expressions such as MOF, MOFs and MOFs. Whether the author refers to the same substance or not puzzled me. The author has a big problem in writing the whole paper.
(4) The content of the author's review does not match the title. It is suggested that the author delete some contents and add corresponding relevant contents.

Reviewer 3 Report

The manuscript submitted by Dr. Li et al. gave a comprehensive review of the application of the metal-organic framework in diabetes. This review is systematic and well-organized, which provides a valuable summary and will guide future studies on this topic. The reviewer only has some minor points:

1)       The grammar of the title is not correct. “its” should be replaced by “their”. It’s actually better to delete “wound healing” from the title because wound healing is already included in the treatment of diabetes.

2)       All figures are too small. It’s very hard to recognize captions in such a small figure.

3)       The “Biomimetic mineralization” should be added to Table 1 as a synthetic method. Refer to https://www.nature.com/articles/ncomms8240.

4)       Insulin is a pH and enzyme-sensitive protein and is usually injected subcutaneously. For section 4. 1 oral insulin delivery, can you elaborate more rationale and mechanism of using MOF as the delivery vehicle and protective shields? Especially how the use of MOF can protect insulins from pH and enzymatic digestion?
